# The North Equatorial Current and rapid intensification of super typhoons

Sok Kuh Kang [1] ✉, Sung-Hun Kim [1], I.-I. Lin [2], Young-Hyang Park [3] ✉, Yumi Choi [4], Isaac Ginis[5], Joseph Cione[6], Ji Yun Shin[1], Eun Jin Kim [1], Kyeong Ok Kim [1], Hyoun Woo Kang [1], Jae-Hyoung Park [7], Jean-Raymond Bidlot [8] & Brian Ward [9] ✉

Super Typhoon Mangkhut, which traversed the North Equatorial Current (NEC; 8–17 °N) in the western North Pacific in 2018, was the most intense Category-5 tropical cyclone (TC) with the longest duration in history−3.5 days. Here we show that the combination of two factors−high ocean heat content (OHC) and increased stratification − makes the NEC region the most favored area for a rapid intensification (RI) of super typhoons, instead of the Eddy Rich Zone (17−25 °N), which was considered the most relevant for RI occurrence. The high OHC results from a northward deepening thermocline in geostrophic balance with the westward-flowing NEC. The stratification is derived from precipitation associated with the Inter-Tropical Convergence Zone in the summer peak typhoon season. These factors, which are increasingly significant over the past four decades, impede the TC-induced sea surface cooling, thus enhancing RI of TCs and simultaneously maintaining super typhoons over the NEC region.

Tropical cyclones (TCs) obtain their energy from the ocean in the form of enthalpy flux[1–3] which is driven in large part by the sea surface temperature (SST) underneath TCs[4,5]. The intensification of TCs has been observed near regions of high SST[6] and statistical and idealized theoretical models have used pre-storm SSTs as a predictor for TC intensity change[7,8].

An important factor controlling the degree of SST cooling is strong vertical mixing that occurs during the TC-ocean coupling processes[9]. During TC's intensification, wind mixing entrains and pumps colder subsurface water into the surface layer[2,3,5,10] which reduces the SST. This cooling effect is a well-known negative feedback to restrain TC's own intensification; the smaller the cooling effect, the more the air-sea heat flux for TC's intensification[10]. As such, ocean heat content (OHC), defined as the upper-ocean heat content relative to the 26 °C isotherm depth (D26)[11] (Methods), maybe a better diagnostic

tool to predict TC-induced cooling than using along-track SST alone[10,12,13]. Over high OHC (deep D26 regions), the TC-induced ocean cooling effect tends to be much smaller, thus more air-sea enthalpy fluxes are available for TC's intensification.

Mesoscale eddies, a transient rotational circulation feature of order 100-200 km in diameter with alternating cyclonic and anticyclonic flows (see Fig. 1a), are known to affect significantly the TC intensity during its life cycle in the western North Pacific[14–19]. Warm eddies, characterized by an anticyclonic circulation and positive temperature/sea level anomalies, cause a deepening of the warm upper layer[20] and thus an increase in OHC, and vice versa for cold eddies. Due to their much deeper D26 and high OHC, warm eddies can effectively restrain the TC-induced ocean cooling effect[6,18,19]. Since the cooling effect controls the air-sea enthalpy fluxes, when TC encounters a warm eddy, it is favorable to develop rapid intensification (RI define as wind

[1]Korea Institute of Ocean Science & Technology, Busan, Korea. [2]Department of Atmospheric Sciences, National Taiwan University, Taipei, Taiwan. [3]Laboratoire LOCEAN/IPSL, Sorbonne Université-CNRS-IRD-MNHN, Paris, France. [4]Korea Institute of Science & Technology, Seoul, Korea. [5]Graduate School of Oceanography, University of Rhode Island, Narragansett, RI, USA. [6]NOAA/AOML Hurricane Research Division, Miami, FL, USA. [7]Department of Oceanography, Pukyong National University, Busan, Korea. [8]European Center for Medium-Range Weather Forecasts, Reading, UK. [9]Air-Sea Laboratory and Ryan Institute, School of Natural Sciences, University of Galway, Galway, Ireland. ✉e-mail: skkang@kiost.ac.kr; young-hyang.park@mnhn.fr; bward@universityofgalway.ie

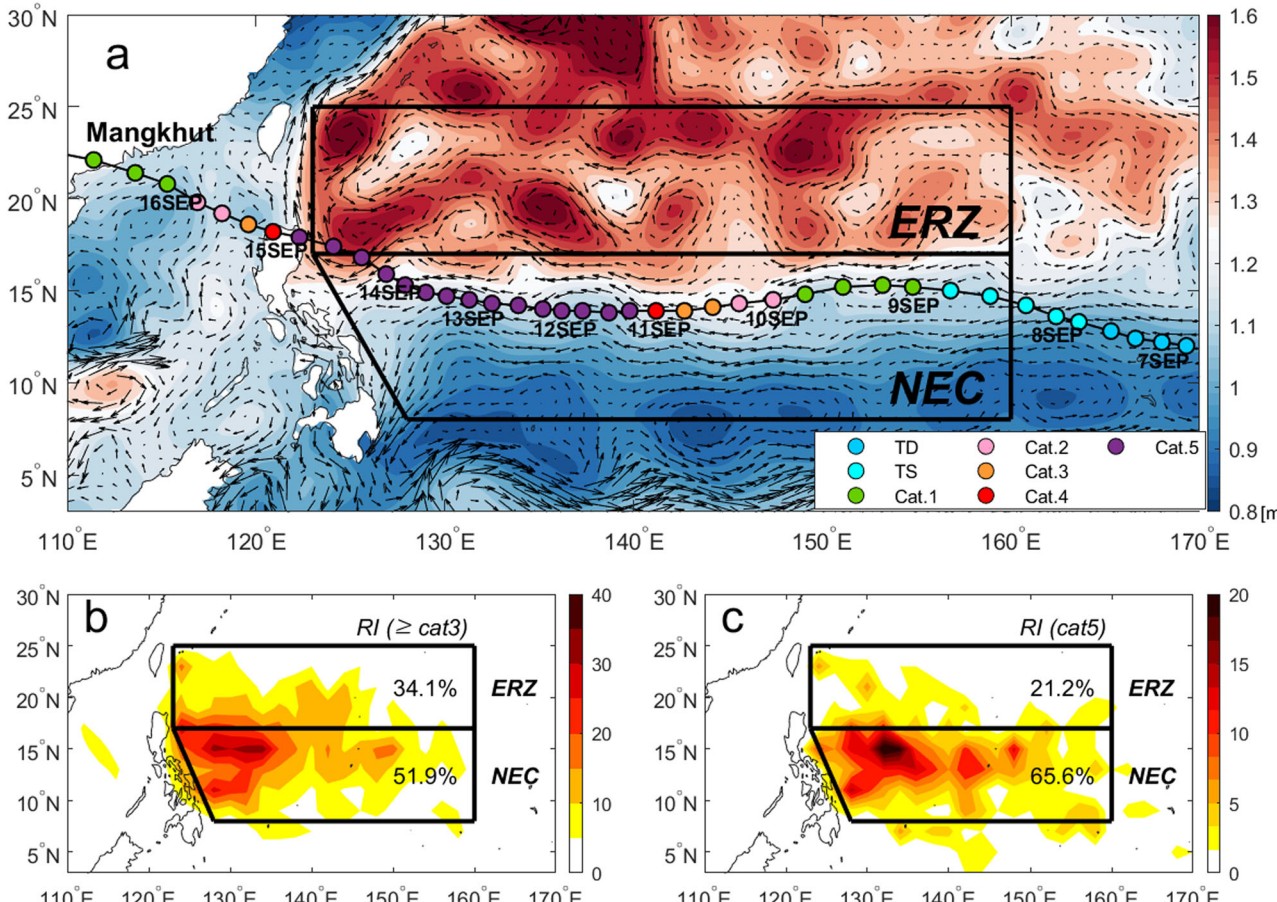

**Fig. 1 | Spatial distribution of satellite-derived absolute dynamic topography on 10 September 2018, and rapid intensification (RI) probability of CAT5 super typhoon from the Joint Typhoon Warning Center (JTWC) best track dataset for 1984–2021. a** Satellite-derived sea surface height overlaid with the Mangkhut track from 7 to 16 September 2018 from the JTWC best track data. The colored closed circles represent the maximum sustained wind speeds corresponding to the tropical cyclone (TC) intensity according to the Saffir-Simpson scale. **b** Spatial distribution of RI events (in %) experienced by all major TCs (≥CAT3; above 50 m s⁻¹). **c** Same as (**b**) but for CAT5 (above 70 m s⁻¹) TCs. For the statistics in (**b**) and (**c**), we included all periods (or events) of RI. Relatively high sea level anomalies are seen to occur in the Eddy Rich Zone (ERZ), with rare warm eddies in the North Equatorial Current (NEC) region.

intensification rate >15.4 m s⁻¹ day⁻¹)[14–17]. Assuming other atmospheric factors (e.g., vertical wind shear, TC translation speed)[21] and sea state[22] are also favorable, encountering warm eddies increases the probability of RI, as the negative feedback effect can be effectively restrained. As a consequence, super typhoons, or Saffir-Simpson scale category-4 (CAT4) or higher (maximum sustained wind greater than 58 m s⁻¹) have been frequently observed in the eddy-rich zone (ERZ) between 17°–25°N[18,19], where mesoscale eddy activity is prevalent, with the frequent occurrence of both cold and warm eddies (Fig. 1a). Notable examples include Super Typhoon Maemi[18] which devastated South Korea in 2003, and 'Killer Cyclone' Nargis[23] which devastated Myanmar in 2008.

In contrast, the area south of the ERZ exhibits little eddy activity and is dominated by a persistent large-scale circulation, characterized by the westward-flowing strong North Equatorial Current (NEC) between 8°–17°N (Fig. 1a). However, analysis of all major TCs (≥CAT3; maximum wind speed greater than 50 m s⁻¹) during 1984–2021, using the Joint Typhoon Warning Center (JTWC) best track database (see Data), reveals a significant number of super typhoons occurring within the NEC region compared to the ERZ (Supplementary Fig. 1a), although mesoscale eddy activity is lacking in the region south of 17°N (Fig. 1a). During the main typhoon season (July–October), eddy activity reaches a maximum in the ERZ[23]. In contrast, during the off-peak typhoon season (December–April), mesoscale eddy activity is at a minimum within the ERZ[23], thus super typhoon activity, while uncommon, is typically observed only within the NEC region (Supplementary Fig. 1b).

For both regions, RI of major TCs most often occurs within the NEC region (52%), compared to 34% within the ERZ (Fig. 1b). The regional contrast is even greater when only CAT5 (maximum wind speed greater than 70 m s⁻¹) TCs are considered (NEC ~ 66%; ERZ ~ 21%; see Fig. 1c). The northern NEC region centered between 14°–15°N and west of 150°E appears to be the most favored region for TC RI for both major TCs and CAT5 TCs (Fig. 1b, c). As indicated similarly in Fig.1b, TCs moving through the NEC have a much higher probability (mean 27.6%) to experience RI than those (mean 6.1%) moving through other regions (Supplementary Fig. 1c). This difference in probabilities was statistically significant based on the Student's *t*-test (*p* < 0.001). Here, we show the key role of the NEC in the RI of super typhoons and their extended persistence in this region by using Mangkhut as a case study.

## Results

### Mangkhut: the longest sustained CAT5 super typhoon over the NEC

Mangkhut was not only the strongest TC in the western North Pacific for 2018, it was the strongest TC globally for that particular year. It moved westward along 14°–17°N over the northern NEC, a region where the least eddy activity occurs, regardless of seasons[23]. On 10 September, Mangkhut's maximum sustained wind increased from 46 to 72 m s⁻¹ over 24 h i.e., its RI phase. Mangkhut traveled westward

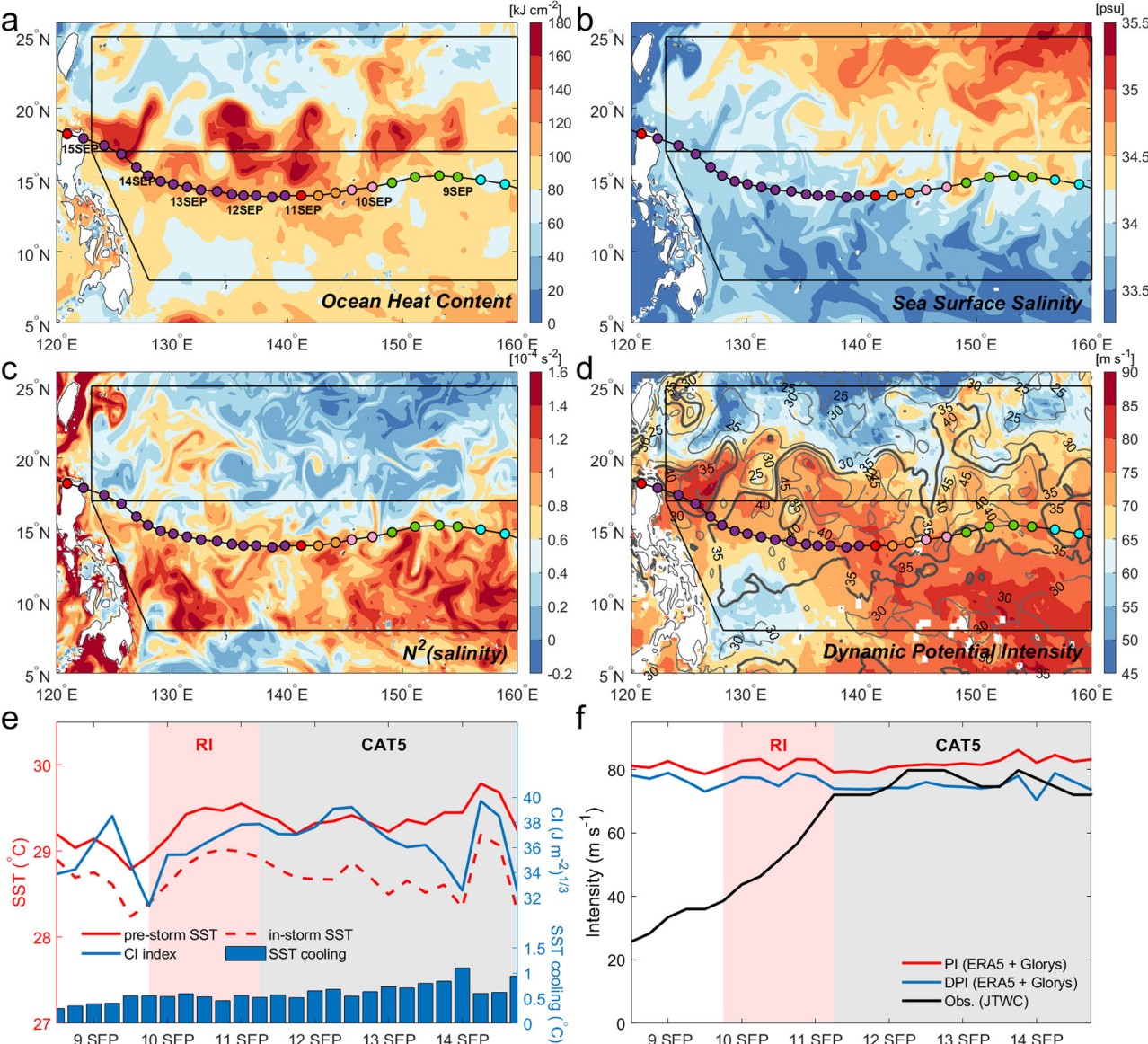

**Fig. 2 | Upper-ocean parameters, sea surface temperature (SST) cooling, and tropical cyclone (TC) wind speed for the Mangkhut case.** Spatial distribution of (**a**) ocean heat content (kJ cm$^{-2}$), (**b**) sea surface salinity (psu, practical salinity unit), (**c**) salinity-induced squared buoyancy frequency ($N^2$) averaged over the first 50-m depth, and (**d**) dynamic potential intensity (DPI; shading in m s$^{-1}$; Methods) and cooling inhibition (CI) index (contours in (J m$^{-2}$)$^{1/3}$; Methods) on 9 September 2018, 1 day before Mangkhut's rapid intensification. **e** Temporal evolution of pre-storm

SST (solid red line), TC-induced cooled SST (discontinuous red line), CI index (solid blue line), and SST cooling (blue bars). **f** Temporal evolution of potential intensity (PI, red), dynamic PI (DPI, blue), and JTWC maximum sustained wind speed (black). In (**e**, **f**) the computation was performed along the TC track using the fifth generation European Center for Medium-Range Weather Forecasts atmospheric reanalysis (ERA5) atmospheric and Global Ocean Reanalysis and Simulation oceanic reanalyses 2 days prior to the TC arrival.

from ~140°E to 122°E (Figs. 1a and 2b) and achieved CAT5 at 06:00 UTC on September 11 and maintained this intensity through 18:00 UTC September 14, making it the longest duration (3.5 days) CAT5 on record.

The cooling inhibition (CI) index (Methods)[24,25] measures the potential energy input required to cool the ocean surface through TC-induced vertical mixing, hence accounts for the pre-storm upper-ocean stratification which resists the heat loss at the surface. The RI region of Mangkhut (150°–140°E) and the area over which Mangkhut maintains CAT5 intensity (140°–122°E) are associated with a large value of CI > 35 (J m$^{-2}$)$^{1/3}$ (Fig. 2d), suggesting that the enhanced upper-ocean stratification (Fig. 2c) by warm/fresher water along the NEC (Fig. 2b, e) exerts a strong inhibition of vertical mixing-driven cooling. Enhanced OHC also prevailed in the NEC region with a maximum centered at its northern boundary (17°N) (Fig. 2a), resulting from the westward-

flowing strong NEC as well as eddy activity near the southern boundary of ERZ, which will be further discussed in climatology. The combination of the NEC-driven high OHC and low salinity-driven upper-layer stratification makes the whole NEC an enhanced region of maximum dynamic potential intensity (DPI, Methods) (Fig. 2d) and explains the persistence of the observed Super Typhoon Mangkhut (Fig. 2f).

Fresher (or low-salinity) water with sea surface salinity lower than 34 psu (practical salinity unit) originated from the rainy Inter-Tropical Convergence Zone (ITCZ) (Fig. 3b). With the seasonal shift of ITCZ's precipitation (Supplementary Fig. 4), fresher water moves northward into the NEC region, coinciding with the peak typhoon season. Dedicated numerical experiments based on the 3D Price-Weller-Pinkel model[26] conducted along the Mangkhut's track show that the fresh water can contribute an additional 19% suppression of SST cooling, a factor which favors TC intensification (Supplementary Fig. 5d, e).

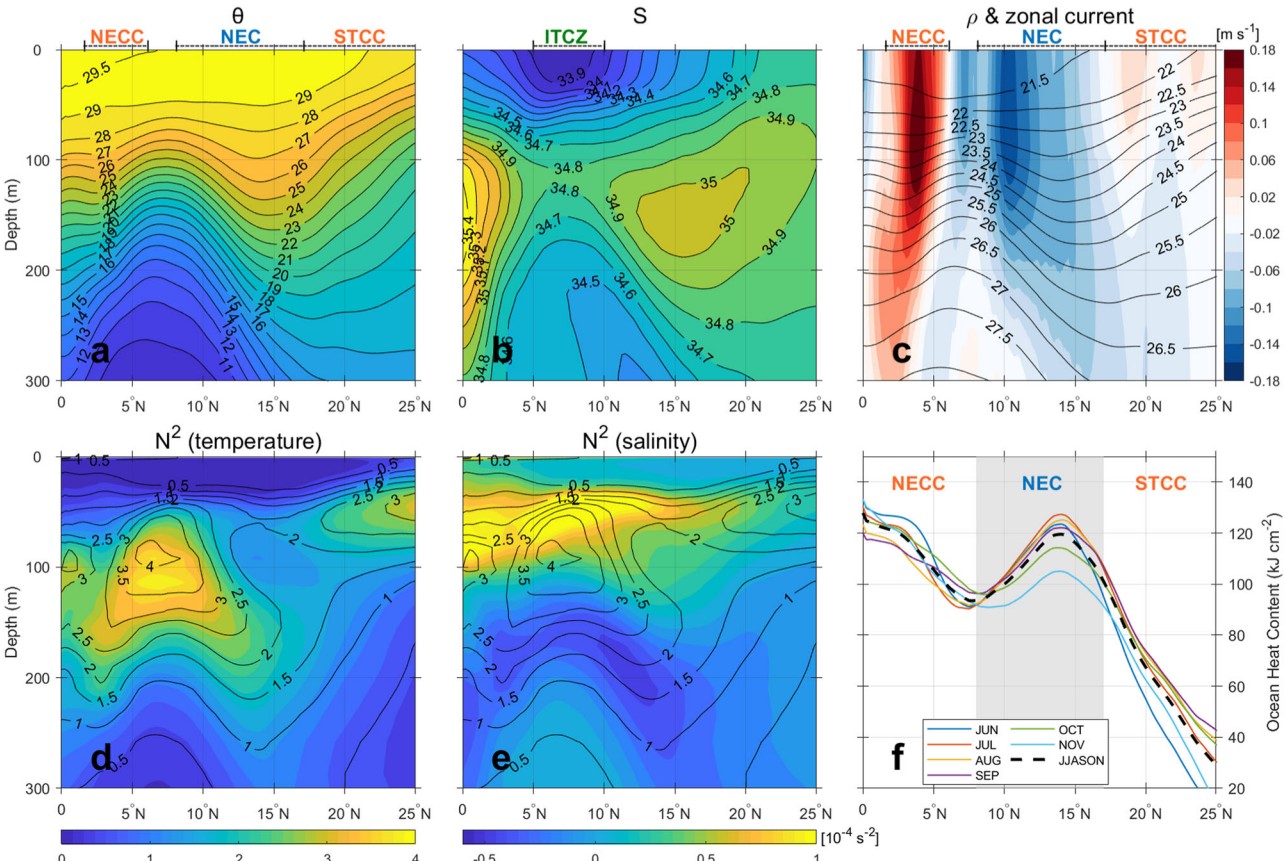

**Fig. 3 | Climatological vertical temperature and salinity sections and ocean heat content across the North Equatorial Current region.** Longitudinal-mean (123°–160°E) meridional sections of (**a**) temperature, (**b**) salinity, and (**c**) density (contour) with overlaid zonal current velocities (shading) between 0° to 25°N, using climatological mean data (1991–2020) from the Ocean Reanalysis System 5 (ORAS5). Corresponding sections of (**d**) temperature- and (**e**) salinity-induced squared buoyancy frequency ($N^2$), and (**f**) ocean heat content. Panels (**a**–**e**) stand for the climatological annual mean sections, while (**f**) shows climatological monthly mean values for different months.

Fresher water (Fig. 2b) combined with elevated pre-storm tropical SST of order 29.0–29.5 °C (Fig. 2e) induces enhanced near-surface stratification (Fig. 2c), resulting in a stronger oceanic inhibition of vertical mixing, which is a very favorable condition for the intensification of TCs (Fig. 2d), assuming other necessary atmospheric conditions also remain favorable. This view is in line with the barrier layer effect for TC intensification[27], as the increased stratification and stability in the salinity-induced barrier layer reduce the TC-induced vertical mixing and SST cooling. Consequently, only limited TC-induced SST cooling between 0.5–0.8 °C, with a peak of order 1 °C, was estimated during Mangkhut (Fig. 2e).

Potential intensity (PI) (Methods) is a theoretical upper bound of TC intensity under given environmental conditions of the atmosphere and the pre-storm SST[28,29]. In contrast, dynamic PI (DPI) (Methods) takes into account the TC-induced SST cooling, significantly improving the PI estimates[30]. The temporal evolution of DPI estimates is in good agreement with the observed JTWC maximum sustained winds of Mangkhut during its long-lasting CAT5, while PI overestimates the observations by about 10 m s$^{-1}$ (Fig. 2f).

While a number of previous work[15,17–19,21] have emphasized the tight link between warm eddies and RI of super typhoons, there were no warm eddies present along Mangkhut's track between 11–13 September (Fig. 1a), thus the existence of the persistent CAT5 can be attributed largely to the OHC maximum centered at 17°N for the Mangkhut case (Fig.2a) compared to 14°N in climatology (Fig. 3f). The longest persistence of Super Typhoon Mangkhut owes to its mostly westward translation, which consistently maintained CAT5 within the northern NEC band. In contrast, the climatological OHC in the ERZ decreases sharply northwards (Fig. 3f), making it more difficult for super typhoons to sustain their intensity for a long time due to the alternating positive/negative OHC anomalies associated with the spatially-varying polarity of transient mesoscale eddy field cut by the TC track. Climatology of best track data shows that no previous CAT5 TCs have been observed to persist for more than 1 day in the ERZ, as was the case with Megi (2010), which maintained a CAT5 for a half day only at 18–19°N[10]. In addition, Maemi (2003) persisted as a CAT5 for less than a day at around ~24°N[18].

It is well known that the vertical wind shear (Methods) greater than a critical value of 10 m s$^{-1}$ is unfavorable TC intensification[31–35]. The pre-storm-mean vertical wind shear during the RI phase of Mangkhut was moderate at 7.12 m s$^{-1}$, whereas during its westward journey as a CAT5 was 7.24 m s$^{-1}$ (Supplementary Fig. 2). The TC translation speed can also exert a significant control on TC intensity by modulating the TC-induced SST cooling, with faster-moving storms tending to lessen the negative SST feedback[36]. The TC translation speed of Mangkhut varied between 5–11 m s$^{-1}$ (Supplementary Fig. 3), considerably more than the 4 m s$^{-1}$ generally accepted as the minimum speed to maintain CAT5[36]. Taken together, this information suggests that the large-scale atmospheric environment was also favorable for the intensification and persistence of CAT5 Mangkhut.

### Enhanced OHC by the North Equatorial Current
The NEC is the largest westward-flowing, wind-driven surface current in the tropical western North Pacific between 8°–17°N and is bounded

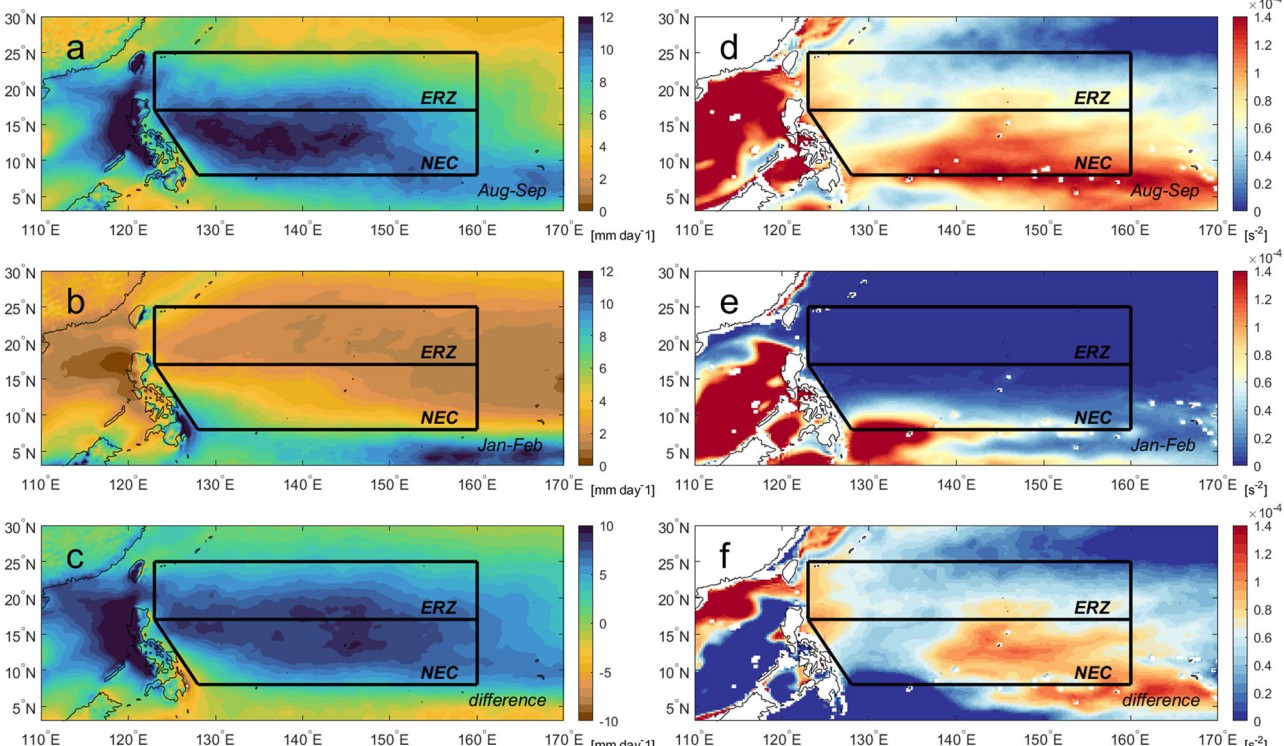

**Fig. 4 | Spatial distributions of climatological seasonal mean total precipitation with its difference (left) and 50 m depth-mean salinity-induced squared buoyancy frequency ($N^2$) and its difference (right) over the North Equatorial Current.** Two-month seasonal mean precipitation (**a**) August-September, (**b**) January-February, (**c**) difference (AS–JF), based upon the fifth generation European

Center for Medium-Range Weather Forecasts atmospheric reanalysis (ERA5). The 50-m depth-mean salinity-induced squared buoyancy ($N^2$) of (**d**) August-September, (**e**) January-February, and (**f**) difference (AS-JF), using climatological data from the ORAS5. ERA5 and ORAS5 are climatological data for precipitation and salinity.

on both sides by two eastward currents: North Equatorial Counter Current in the south and the Subtropical Countercurrent in the north[37–39] (Fig. 3c). The 30-year (1991–2020) climatology of zonal-mean (126°–160°E) meridional sections of water properties (Fig. 3a–c) shows that the upper-layer thermocline is shallowest in the ITCZ at 8°N, which deepens gradually both to the south and to the north until encountering a local maximum at 14°N before shoaling again to the north in the ERZ. The meridional gradient of upper-layer isopycnals (23.0–26.5 kg m$^{-3}$) appears in geostrophic balance with the upper-mentioned system of three currents (Fig. 3c): northward deepening of isopycnals by the strong westward NEC between 8–17°N, in contrast to their northward shoaling by both the strong eastward North Equatorial Counter Current south of 8°N and the weak eastward Subtropical Countercurrent in the ERZ north of 17°N.

The northern flank of the NEC between 123° and 160°E exhibits the deepest thermocline, with the 26 °C isotherm (used for the computation of OHC) deepening from a minimum of 90 m at 7°N to a maximum of 130 m at 14°N (Fig. 3a) due to the westward-flowing NEC. More specifically, deepening of thermocline results in OHC values > 100 kJ cm$^{-2}$ over the NEC latitude range (8°–17°N), with a peak of 120 kJ cm$^{-2}$ at 14°N (Fig. 3f). For the ERZ north of 17°N, however, the climatological OHC drops sharply below 100 kJ cm$^{-2}$, reach a value as low as 40 kJ cm$^{-2}$ at 25°N. Consequently, in the ERZ the presence of warm eddies is indispensable for compensating the climatologically low OHC, and creates a critical value of in-situ OHC for the accommodation of super typhoons. This process is not required for the NEC region.

In addition, rain-induced fresh water from the ITCZ enhances the stratification in the upper 50 m in the NEC region (Fig. 3b, e), suggesting the presence of a barrier layer[27]. The ITCZ undergoes seasonal migration from the equatorial region in winter northwards into the NEC region in summer, with the peak summer precipitation

dominating over winter precipitation (Fig. 4a–c). This feature results in the salinity-driven squared buoyancy $N^2$ (indicative of stratification) in summer overwhelming winter $N^2$ in the NEC region (Fig. 4d, e). The ITCZ freshening forms a strong barrier layer (Supplementary Fig. 4d, f), contributing positively to the CI index by 7.5–12.5% in the NEC region compared to that without the freshening effect (Supplementary Fig. 6), implying that it is a significant secondary factor in the NEC for the RI of TCs. Theoretical cooling by CAT5 super typhoons translating at 6–7 m s$^{-1}$ is estimated to be less than 0.8–0.9 °C in the NEC region (Supplementary Fig. 7a; Methods), in agreement with the observed cooling during the mature phase of Mangkhut (Fig. 2e). In addition, summer freshening occurs just on time to coincide with the peak typhoon season, leading to RI of TCs, with heat flux over 800 Wm$^{-2}$ (Methods) similar or higher than the reported magnitudes[2,15] during RI.

Consequently, the combined effect of the ITCZ freshening-induced enhanced stratification in the upper layer (though a secondary factor) together with the deepest thermocline due to the westward-flowing strong NEC (as a primary factor) makes the whole NEC region a hot spot for accommodating the strongest TCs. This finding is consistent with the predominant occurrence of RI events for super typhoons in the northern NEC region (Fig. 1b, c).

## Discussion

We have shown that the upper-ocean stratification in the NEC region naturally features the highest OHC due to the northward thermocline deepening in geostrophic balance with the strong westward-flowing NEC. The highest OHC combined with enhanced subsurface stability due to the fresher water supply by the summer northward migration of the ITCZ makes the northern NEC region between 14°–15°N west of 150°E the most favorable corridor for the intensification and persistence of super typhoons (Fig. 1b, c).

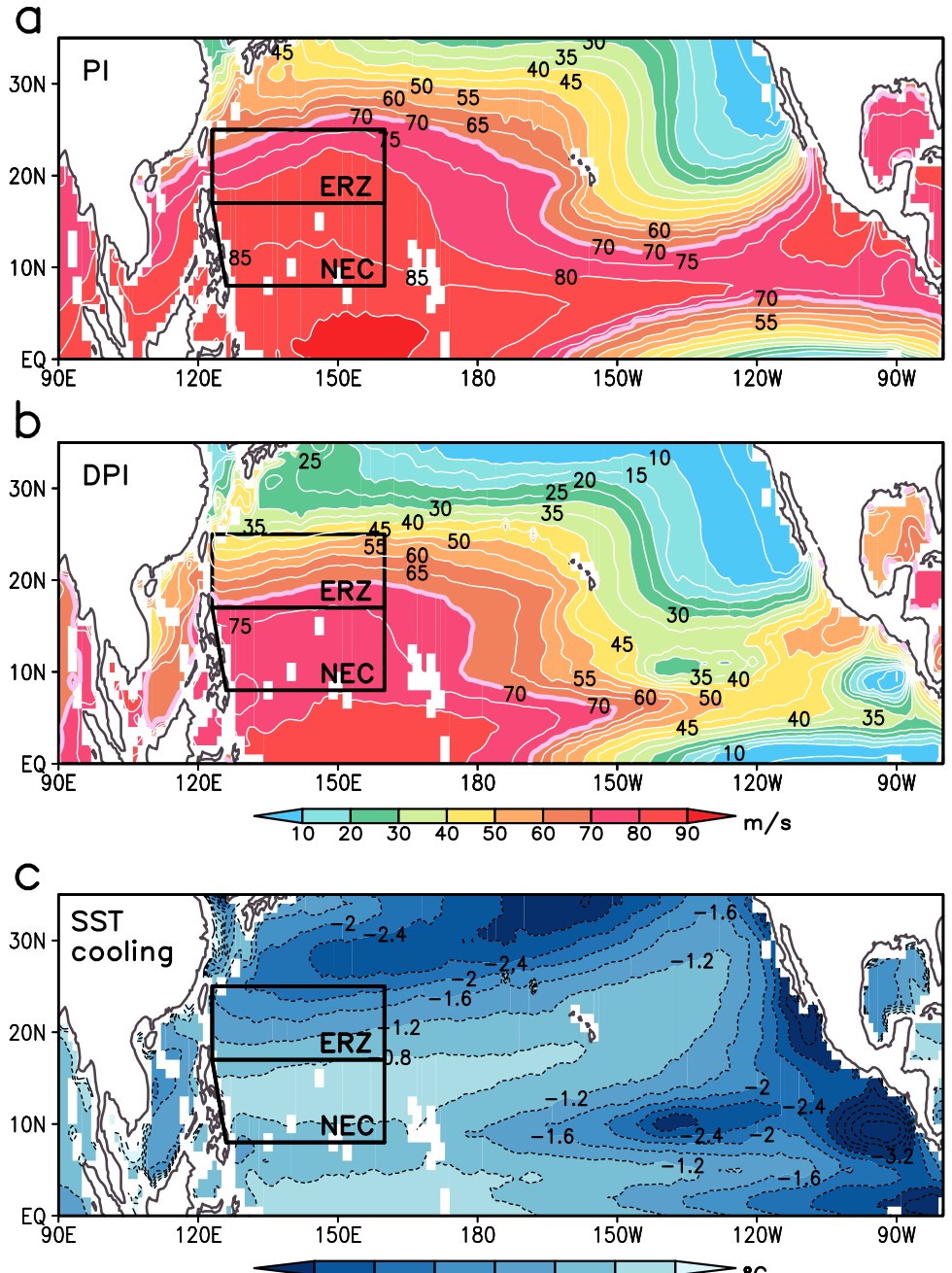

**Fig. 5 | Climatological (1991–2020) Potential Intensity (PI) and Dynamic Potential Intensity (DPI) under given thermodynamic environment of the atmosphere and the upper ocean.** Spatial distributions of (**a**) PI, (**b**) DPI, and (**c**) sea surface temperature cooling (shading/contours, see Methods) computed from the season-mean (June–November) European Center for Medium-Range Weather Forecasts (ECMWF) reanalyses (the fifth generation ECMWF atmospheric reanalysis (ERA5) and Ocean Reanalysis System 5 (ORAS5)) and DPI from (**b**). All grid points (0.25° × 0.25°) were forced by the typical CAT5· tropical cyclone (TC) characteristics inspired from Mangkhut: maximum sustained wind speed of 70 m s$^{-1}$; radius of maximum wind of 35 km; TC translation speed of 5 m s$^{-1}$. The two black boxes in the western North Pacific represent the Eddy Rich Zone and North Equatorial Current regions, respectively.

To understand the key role that the NEC region plays in RI of super typhoons in a global perspective, Fig. 5 compares the climatological PI and DPI in the North Pacific south of 35°N. The area with PI values over 70 m s$^{-1}$ (a lower bound of CAT5 TC) extends up to 25°N, encapsulating most of the ERZ (Fig. 5a), while the DPI zone contracts into the NEC region (Fig. 5b). The difference between these two indices indicates the contribution from the TC-induced negative SST feedback, which accounts for less than 10 m s$^{-1}$ in the NEC region, compared to a sharp increase of up to 30 m s$^{-1}$ in the ERZ. The associated SST cooling is minimal and less than 0.8 °C in the NEC region (Fig. 5c), similar to the

climatology in the literature[25] and our climatology (Supplementary Fig. 7a), as well as to the Mangkhut observations (Fig. 2e).

These climatological results are consistent with the Mangkhut case (Fig. 2), which confirms that the westward NEC-associated high OHC and the ITCZ-related haline stratification are optimally conducive for inhibiting TC-induced SST cooling, thus systematically favoring the intensification and persistence of super typhoons in the NEC region. However, Mangkhut is not an isolated case, as we have also observed in May 2023 another long-lasting (2.7 days) super typhoon in the NEC region, even during the pre-typhoon season (Supplementary Fig. 8). In

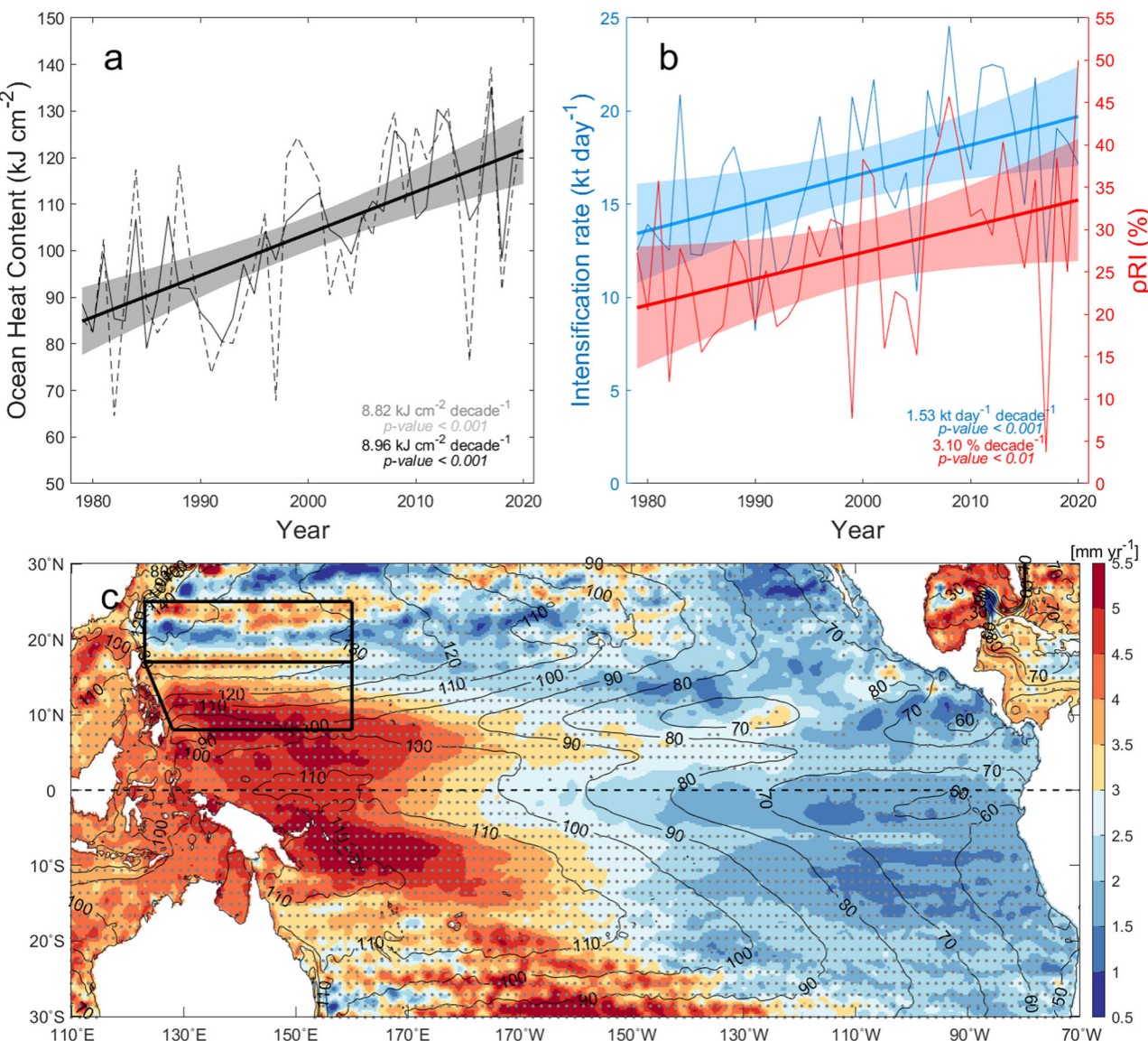

**Fig. 6 | Increasing ocean heat content (OHC) and the probability of rapid intensification (RI) events in the North Equatorial Current region, along with the climatic warming of the western North Pacific.** Time series of (**a**) seasonal mean OHC, (**b**) intensification rate (blue line; kt per day), probability of RI event (red line; %), and their linear regression (thick lines) in the North Equatorial Current region. **c** Map of sea level trend from altimetry for 1993–2021 (Methods). In (**a**) the solid black line represents OHC time series without ENSO signals, which have previously been removed by subtracting the ENSO-regressed time series from the original time series (discontinuous line) of European Center for Medium-Range Weather Forecasts Ocean Reanalysis System 5 (ORAS5; see Methods). The shaded area in (**a**) and (**b**) indicates a 95% confidence interval. In (**c**) statistically significant trends based on the nonparametric Mann–Kendall test are marked by black dots ($p < 0.05$), and black contours indicate the mean sea level field (cm) for 1993–2020.

effect, Mawar intensified from CAT3 to CAT5 on 25 May soon after it changed its translation direction westward from NW to WNW at 14°N, 144°E. It then maintained its CAT5 intensity for 1.5 days (25–26 May) while it moved westward in the northern part of the NEC region (14°–16°N), before turning northward and gradually weakening to become CAT2 in the ERZ north of 17°N after 28 May. Upper-ocean stratification conditions during the CAT5 phase of Mawar on 26 May were very favorable (OHC - 130 kJ cm⁻² and SST cooling -0.5 °C), although the low salinity inflow from the ITCZ was not significant (Supplementary Fig. 8). These upper-ocean conditions during the CAT5 phase of Mawar are contrasted with significantly less favorable conditions during its weakening phase on 27 May (OHC < 70 kJ cm⁻² and SST cooling >1.5 °C). Note also that the atmospheric conditions were also favorable during the CAT5 phase of Mawar, with VWS less than 5 m s⁻¹ and the translation speed greater than 5 m s⁻¹ (Supplementary Fig. 9). Therefore, Mawar supports our findings from

Mangkhut that super typhoons are likely to both rapidly intensify and persist whenever they travel westward within the NEC region under favorable atmospheric conditions.

We have emphasized the utmost importance of the NEC as a preferred region for rapid intensification and long-lasting persistence of maintaining super typhoon strength in the western North Pacific, which has hitherto received little attention. First identified from the analysis of Mangkhut, the unexpected contribution of the NEC to the RI of super typhoons is found to be a quasi-permanent climatic feature. Of particular importance in this regard is the increase in OHC over the NEC at a rate of -9 kJ cm⁻² decade⁻¹ over the last 4 decades (Fig. 6a), which far exceeds the global (1.32 kJ cm⁻² decade⁻¹) and all-Tropics (2.47 kJ cm⁻² decade⁻¹) trends (Supplementary Fig. 10). Furthermore, the mean intensification rate of TCs in the NEC region has increased by approximately -1.5 kt day⁻¹decade⁻¹, which equates to a probability of RI of 3.1% per decade (Fig. 6b).

Previous work[40–42] as well as Fig. 6c show that the overall increase in OHC and RI in the NEC region arise from warming (upward trend in sea level) of the western North Pacific. The El Nino-Southern Oscillation (ENSO) may also play a role in the interannual variability of OHC and RI in the NEC region[43], which needs further analysis, although removing the ENSO signal is unlikely to affect the long-term trend of OHC (Fig. 6a).

## Methods

### Environmental conditions from satellite and reanalysis data
The daily absolute dynamic topography, produced by the satellite altimeter data with a spatial resolution of 0.25° × 0.25°, was used for the subsurface thermal condition and the existence of eddies[44]. It was obtained from the Copernicus Marine Environment Monitoring Service (http://marine.copernicus.eu). For the estimation of atmospheric conditions for Typhoon Mangkhut, we used the 6-hourly fifth generation European Center for Medium-Range Weather Forecasts (ECMWF) atmospheric reanalysis (ERA5) with a spatial resolution of 0.25° × 0.25° and 37 vertical levels[45]. To analyze ocean states, we used vertical profiles of water temperature and density from the Global Ocean Reanalysis and Simulation (see GLORYS in Methods)[46]. The daily GLORYS12V1 was used with a standard regular grid with 1/12° (approximately 8 km) interval and 50 standard levels. The data are available at https://data.marine.copernicus.eu/products. To investigate the impact of climatological atmospheric conditions and ocean states on the TC intensification, the monthly ERA5 and Ocean Reanalysis System 5 (see ORAS5 in Methods)[47] with 75 vertical levels were utilized with a horizontal grid of 1°.

### JTWC best track data
For tracking Mangkhut, we used the best track data from the Joint Typhoon Warning Center (JTWC).

### Satellite-based surface ocean environmental data
Satellite-based data were used for examining the surface environmental conditions associated with the spatio-temporal evolution of Super Typhoon Mangkhut (2018). Optimally-interpolated microwave and infrared SST data were used[48]. Daily absolute dynamic topography and sea level anomaly data with a spatial resolution of 0.25° × 0.25° were utilized for describing the surface circulation and mesoscale eddies[44]. To identify the effect of low salinity water from the ITCZ, daily sea surface salinity data interpolated into a 0.25° grid with an accuracy of 0.2 psu were used[49].

### GLORYS12V1 model data
GLORYS12V1 is the CMEMS global ocean eddy-resolving (1/12° horizontal resolution and 50 vertical levels) reanalysis covering the altimetry era (1993–2018)[46]. It is largely based on the current real-time global forecasting CMEMS system (Marine.Copernicus.EU). The ocean model component is the NEMO[50] driven at the surface by the ECMWF ERA-Interim reanalysis with a horizontal resolution of ~80 km[51]. The global ocean data were displayed on a standard regular grid with 1/12° (approximately 8 km) intervals and 50 standard levels.

### ORAS5 reanalysis data
ORAS5 is global ocean and sea-ice reanalysis data produced by the ECMWF OCEAN5 analysis-reanalysis system with a spatial resolution of 0.25° × 0.25° and 75 vertical levels[47]. The ORAS5 system comprises 5 ensemble members and covers from 1979 onwards. This system uses the NEMO ocean model and the NEMOVAR ocean assimilation system, which uses the 3D-var FGAT assimilation technique. The ORAS5 data is constrained by global atmospheric reanalysis and observation data (SST, salinity, sea-ice concentration, sea-level trend, and climatological variation of the ocean mass).

### ERA5 reanalysis data
ERA5 provides hourly and monthly estimates of the state of the atmosphere on pressure and single levels with the regular lat-lon grid of 0.25°[45]. The data are produced by the ECMWF combining model data with observations and available from 1940 onwards. A 10-member ensemble at 3-h intervals is used to produce an uncertainty estimate.

### Ocean heat content (OHC)
According to Leipper, D. F. & Volgenau[11], the pioneer of this parameter, the sea surface loses heat to the atmosphere in a TC, leading to cooling by turbulent convection through the mixed layer. Considering the air temperature of 26 °C, which is a dew-point air temperature in the tropics, the sea-to-atmosphere heat transfer would continue until the mixed layer temperature drops to 26 °C. This is the maximum heat utilized by a TC, initially referred to as 'TC heat potential', equivalent to OHC, which is computed based on the following formula from ref. 11.

$$\text{OHC} = \int_{-H26}^{0} Q(z)dz, \text{ with } Q(z) = \rho c_p(T(z) - 26) \tag{1}$$

where H26 is the depth of the 26 °C isotherm, $\rho$ is water density (~1025 kg m$^{-3}$), $c_p$ is the specific heat at constant pressure (~3850 J (kg °C)$^{-1}$), and $T(z)$ is water temperature in °C at a depth $z$. OHC time series figures (Fig. 6a and Supplementary Fig. 10) are plotted using the original time series of ORAS5 (Methods).

### Cooling inhibition index (CI)
Vincent et al.[25] suggested an idea to assess oceanic control on the amplitude of sea surface cooling induced by TCs. Given a pre-storm upper-ocean density profile, one can calculate the potential energy increase $\Delta E_p(\Delta T)$ necessary to produce a given $\Delta T$ cooling assuming heat conservation and a perfectly homogeneous mixed layer after the mixing by the typhoon. The larger $\Delta E_p$, the more energy has to be injected into the ocean to produce TC-induced cooling. This is to characterize the propensity of the pre-storm ocean state to yield a weak or strong surface cooling in response to a surface kinetic energy input.

$$\Delta E_p(\Delta T) = \int_{h_m}^{0} \left(\rho_f - \rho_i(z)\right) gz dz \tag{2}$$

where $\rho_i$ is the initial unperturbed profile of the density, $\rho_f$ is the homogeneous final density profile, $g$ is the acceleration due to gravity, and $z$ is ocean depth. Following Vincent et al.[25], we adopted the Cooling Inhibition index (CI) as the cube root of the necessary potential energy to induce a 2 °C SST cooling:

$$\text{CI} = \left[\Delta E_p(-2°\text{C})\right]^{1/3} \tag{3}$$

The GLORYS12V1 output was used for the density profile needed to calculate the CI.

### Potential intensity (PI)
Emmanuel[29] demonstrated that potential intensity (PI) is a useful statistical analysis of TC intensity because it provides the upper limit TC can achieve under the thermodynamic state of the atmosphere and sea surface. PI[27] is expressed as

$$V_{\max} = \alpha \sqrt{\left\{\left(\frac{C_k}{C_D}\right)\left(\frac{T_s}{T_o}\right)\left(\text{CAPE}^* - \text{CAPE}\right)_{\text{RMW}}\right\}}, \tag{4}$$

where $\alpha$ is the factor to reduce gradient wind by friction for 10-m wind as 0.8 by default, $C_k/C_D$ is the ratio of the exchange coefficient for enthalpy ($C_k$) to the drag coefficient ($C_D$) as 0.9 by default, replaced by

1.0 in this study, $T_s$ is the SST, $T_o$ is the outflow temperature, CAPE* is the convective available potential energy of saturated air lifted from sea level to the outflow level, and CAPE is that of the environment near the radius of maximum wind (RMW). The 6-hourly ERA5 and daily GLORYS data 2 days before the TC arrival were used to calculate the 6-hourly PI. For the climatological PI, monthly ERA5 and ORAS5 were used.

## Dynamic potential intensity (DPI)
Balaguru et al.[30] suggested the dynamic PI (DPI) reflect both ocean stratification and TC state by replacing SST with the vertically averaged seawater temperature $T_{dy}$ over the mixing length $L$ as

$$T_{dy} = \frac{1}{L} \int_0^L T(z)dz, \qquad (5)$$

The mixing length forced by the TC circulation is calculated as

$$L = h + \left(\frac{2\rho_o u_*^3 t}{\kappa g \alpha}\right)^{\frac{1}{3}}, \qquad (6)$$

where $h$ is the initial mixed layer depth defined as the depth where a potential temperature decreases by 0.2 °C relative to its value at 10 m depth, $\rho_o$ is the seawater density, $u_*$ is the friction velocity in the mixed layer calculated from the maximum sustained wind of TC obtained using a drag coefficient for high winds[52,53], air density of 1.2 kg m$^{-3}$ [30], $\kappa$ is the von Kármán constant of 0.4, $g$ is the gravitational acceleration, and $\alpha$ is the rate of increase of potential density with depth below the mixed layer. $t$ is the mixing time interpreted as the duration of the TC passage across the mean radius of maximum wind. The 6-hourly ERA5 and daily GLORYS data were used to estimate environmental states associated with Typhoon Mangkhut. To calculate the spatial distribution of DPI on 9 September 2018, 2 days before Mangkhut reached its CAT5 intensity, all of the grids were forced by Mangkhut with intensity of 75.45 m s$^{-1}$, translation speed of 6.54 m s$^{-1}$, and radius of maximum wind of 34.4 km which are the mean values over the period of sustained CAT5 (0600 UTC 11–1800 UTC 14 September 2018; Supplementary Fig. 3). The temporal evolution of DPI over the typhoon center during the period of 1200 UTC 08–1800 UTC 14 September 2018 was calculated from the JTWC best track data for Mangkhut, with a fixed radius of maximum wind of 35 km.

## Vertical wind shear (VWS)
The magnitude of vertical wind shear (VWS) is defined as below.

$$\sqrt{(U_{200hPa} - U_{850hPa})^2 + (V_{200hPa} - V_{850hPa})^2}, \qquad (7)$$

where $U_{200hPa}$ and $V_{200hPa}$ ($U_{850hPa}$ and $V_{850hPa}$) are horizontal winds at 200 (850) hPa. The VWS along the track of Mangkhut was calculated by averaging within a radius of 200 km relative to the TC center 2 days before the TC arrival, although the results are not sensitive to lags 0–2 days (see Supplementary Fig. 2c).

## TC-induced Sea Surface cooling (d$T$)
CAT5 super typhoon with a translation speed of 6–7 m s$^{-1}$ corresponds to wind power index WPi = 4.1. (Fig. 4b of ref. 25). Following Vincent et al.[25], d$T$ is depend on WPi and cooling inhibition index CI from a least square fit. For the fixed WPi = 4.1, TC-induced cooling d$T$ is a function of CI only (Fig. 7b of ref. 25). d$T$ is estimated from a 2nd order polynomial fitting between d$T$ and CI (Fig.7b of ref. 25) as:

$$dT = 5.5578 - 0.2233 \times CI + 0.0025 \times CI^2 \qquad (8)$$

Latent Heat Flux during Rapid Intensification in the NEC.

Following previous work[2], the latent heat flux bulk formula is defined as:

$$LHF = \rho\, L_e\, C_e\, U10\, (Qs - Qa), \qquad (9)$$

where $\rho$ = 1.12 kg m$^{-3}$ is the air density, $L_e$ = 2.5 × 10$^{-6}$ is the latent heat of vaporization, $C_e$ = 1.4 × 10$^{-3}$ (HWRF, 2017) is turbulent exchange coefficient, $U10$ is 10-m wind speed, $Q_s$ is saturation specific humidity, and $Q_a$ is specific humidity.

The latent heat flux estimated from (9) using representative parameters for RI in the NEC region is usually around 838-915 Wm$^{-2}$, which is similar to or higher than the reported enthalpy flux (600–900) Wm$^{-2}$ for RI[2,15]. This order of magnitude of latent heat flux confirms that RI can easily occur in the NEC region due to its favorable upper-ocean stratification described in the text.

### Sea Level Rise Trend in the NEC
Sea level data from satellite altimetry is available from NOAA (https://cds.climate.copernicus.eu/cdsapp#!/dataset/satellite-sea-level-global?tab=overview). The tropical western Pacific basins are among the steepest sea level rising areas in the world's oceans (Fig. 6c). Sea level change in the western North Pacific is due mostly to the upper-ocean thermosteric component[40], implying that it is a good proxy of OHC change. Note that the sea level rising trend in the NEC region during the past three decades is much stronger than that in the ERZ by a factor of 2.

### (ARGO I, II) data
ARGO I data are available by the international Argo Program, measured by a traditional Argo float[54] with one day interval and ARGO II data are measured by a similar type of float with three-hour interval data, during May 25 to 29, 2023. They were used for plotting Supplementary Fig. 8.

## Data availability
These satellite-based sea surface temperature and salinity data were provided by the Remote Sensing Systems in Santa Rosa, California (http://www.remss.com). The satellite altimeter data were obtained from the Copernicus Marine Environment Monitoring Service (https://cds.climate.copernicus.eu/cdsapp#!/dataset/satellite-sea-level-global?tab=form). Tropical cyclone information is obtained from the Joint Typhoon Warning Center (JTWC), available at http://www.metoc.navy.mil/jtwc/jtwc.html. GLORYS12V1 model data are available at. https://data.marine.copernicus.eu/product/GLOBAL_MULTIYEAR_PHY_001_030/description. ORAS5 reanalysis data are available at https://cds.climate.copernicus.eu/cdsapp#!/dataset/reanalysis-oras5?tab=overview. ERA5 reanalysis data are available at https://cds.climate.copernicus.eu/cdsapp#!/dataset/reanalysis-era5-pressure-levels?tab=form. The ARGO I data are available at https://doi.org/10.17882/42182#5906387 and ARGO II data are available upon request.

## Code availability
Codes used to produce results and figures were obtained using Matlab and Grads software package. All codes used in this study can be obtained from the corresponding author upon request.

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

## Acknowledgements

This research was supported by Korea Institute of Marine Science & Technology Promotion (KIMST) funded by the Ministry of Oceans and Fisheries (20220566). B.W. was supported with a research grant from Science Foundation Ireland (SFI) under Grant Number 13/RC/2092_2 as part of the iCRAG centre.

## Author contributions

S.K.K. designed the study and wrote the initial draft of the manuscript. S.H.K. conducted the analysis for Figs. 1b, c, 3, 4, and 6, discussed the major results. I.I.L. raised the necessity of investigating salinity effect in some detail, estimating salinity contribution for Mangkhut and Supplementary Fig. 5. Y.H.P. revised the manuscript in form of more physical interpretation, and let this study examine SST cooling in terms of PI and DPI yielding Fig. 5, as well as revising original Fig. 2. and suggestion to include Fig. 6c. Y.C. analyzed some figures of Fig. 2 and prepared for Fig. 5 with analysis of PI and DPI. I.G. and J.C. read through the manuscript and give some comments, including SST cooling. J.Y.S. analyzed statistics of major TCs for Supplementary Fig. 1 and analyzed data for SST cooling shown in Supplementary Figs. 6 and 7, along with estimate of heat flux during rapid intensification in NEC. E.J.K. prepared for initial version of Fig. 1a and analyzed argo data for Supplementary Fig. 8. K.O.K. discussed the rapid intensification of Mangkhut and H.W.K. contributed to collecting argo/arbo profile data and discussed the rapid intensification. J.H.P. computed cooling inhibition index (CI) to prepare for CI at Fig. 2d, e, contributing to preparing for Supplementary Figs. 7 and 8. J.R.B. initially analyzed atmospheric condition during the Mangkhut propagation and rapid intensification, in order to understand the process of rapid intensification, and discussion for long term trend of RI typhoon in warming world. B.W. revised and edited the initially submitted and revised version of manuscript, along with response to comments by reviewers.

## Competing interests

The authors declare no competing interests.
