## [Peer Review File · Nature Communications]

The North Equatorial Current and Rapid Intensification of Super TyphoonsREVIEWER COMMENTS

Reviewer #1 (Remarks to the Author):

Comments on “The North Equatorial Current and Rapid Intensification of Super Typhoons” by Kang et al.

This study explores the cause of typhoon Mangkhut in 2018. The authors especially emphasize the role of North Equatorial current. They also analyzed the role of ITCZ induced freshening. Overall, this work is interesting. I think most of the analysis focuses on the happening of the long-lasting super typhoon Mangkhut. So, it is more like a case study. But the most intrinsic reason for this is still not clear based on the current evidence. The main reason I think is too much analysis is about the climatology. I will show some examples below. Thus, I think the authors need further investigation into this particular case, that is the typhoon Mangkhut. Please see specific comments below.

Specific comments:

1. The TC intensification is closely related to the upper ocean heat supply of course, especially for the typhoon Mangkhut, which moves along the NEC region. So, it will be not surprise that the OHC in NEC region plays critically important role in sustaining the strong intensity of Mangkhut. The authors indeed show several largescale conditions that favorable for the intensification, like the large OHC, moderate vertical wind shear and freshwater induced by ITCZ shift. But only Figure 2 is about this particular case. Figures 3 and 4, which is also important for TC intensification, are both on climatological perspective. Although the authors have analyzed in many details, it is still unclear how these climatological variables impact the particular case Mangkhut. For example, in L39-41, the ITCZ northward migration is seasonal march, but why only Mangkhut sustained its high intensity, while other TCs doesn't. This issue exists in the analysis of precipitation, salinity, buoyancy et al. I suggest the authors to carried out these analysis for Mangkhut as in Figure 2.
2. The atmosphere changes much faster than the ocean. Thus, I concern that in the analysis of vertical wind shear around L149-158, whether it is still proper to analyze the pre-storm VWS as that of the OHC. The VWS is likely to change with time during Mangkhut. So, how does VWS change during all life cycle of Mangkhut should be investigated.
3. Figure 6. The authors show the long-term trend of the OHC and TC intensification rate over the NEC region. Actually, many study show that the overall OHC in the WNP has increased not limited to NEC region. So, the figure 6a only shows a part of the overall warming pattern of the WNP. This should be acknowledged in the manuscript and also should be discussed on the potential relationship between NEC warming and the overall warming pattern. Another issue is that ENSO may plays a role in the long-term trend of OHC. Thus, it's signal should be removed.
4. The following study is quite relevant to this study, which found that a type of fast decaying El Nino events leads to westward migration of RI events by affecting the NEC. The authors should not miss it in reviewing the related background:

Guo Y. P., and Z. M. Tan, 2018: Westward migration of tropical cyclone rapid-intensification over the Northwestern Pacific during short duration El Niño, *Nature Communications*, 9, 1507, DOI: 10.1038/s41467-018-03945-y.

5. Figure 6c: The significance. The authors say shading means significant. Does it mean the trends of sea level over the whole Pacific is statistically significant?

Reviewer #2 (Remarks to the Author):

Reviewer: Sam Hardy

Overview

The authors present an interesting and informative analysis of the role of the North Equatorial Current region on the rapid intensification of super typhoons in the western North Pacific. Data are presented impressively both for the case study of Typhoon Mangkhut (2018) and over the longer period of study used for climatological analysis. The interesting findings represent a positive contribution to the field, particularly the analysis on the salinity and ocean stratification, and I congratulate the authors on putting together this piece of work. However, there are several aspects of the discussion that need improving before the paper is suitable for publication, which I have outlined below. My recommendation is Major Revisions.

General comments

- Reduce the number of acronyms throughout the paper. It's difficult for the reader to keep track of them after a certain point.
- Given the central importance of mesoscale eddies to your analysis, you need to introduce the topic in more detail at the start of the paper. It currently feels like you're assuming that the reader already knows what all these concepts mean.
- Linked to the previous point, how does mesoscale eddy activity contribute to the rapid intensification of tropical cyclones? You need to better introduce the established literature on this topic to give the reader a more solid understanding by the time you get to your specific analysis.
- Introduce the "eddy-rich zone" and the "North Equatorial Current" region to the reader in more detail early in the paper and discuss their relevance to tropical cyclone intensification. Much of the analysis that comes later would be easier to follow if the introduction of these terms was more robust.
- In the abstract you mention the intensification of super typhoons in a warming climate. However, you do not discuss this point any further in the manuscript, only touching on the topic again right at the end of the paper (L248-250). Either expand this part of your analysis or modify the abstract to avoid confusing the reader.
- You need more robust discussion of the relationship between upper-ocean stratification and tropical

cyclone intensification. Making this change would really add depth to your analysis.

- Create a stronger link between your analysis of Typhoon Mangkhut and your overall conclusions, for example in the Summary and Discussion section. It almost feels like the paper contains two separate bits of analysis, which are not properly linked together. Along the same lines, the paper should end with a more confident and conclusive statement on your findings.

Specific comments

- L31-45. Reduce the number of acronyms in the abstract.
- L33. Avoid using “this” at the start of the sentence without telling the reader what you’re referring to (also L128, L180, L194, L218, L237).
- L35. Define “freshening” here (and again in the main text).
- L31-45. Discuss how the occurrence of Typhoon Mangkhut relates to your results in a bit more detail. You only mention Mangkhut once (in the first sentence).
- L53. Spell acronyms out fully when they occur at the start of a sentence (e.g. “OHC”).
- L56. What do you mean by “warm-core eddies”? You introduce this term in conjunction with a single reference but don’t go into any more detail.
- L57. Be more specific about “other atmospheric factors”.
- L77. Spell out “NEC” fully; you have introduced the term in the abstract and the caption of Figure 1, but not in the main text.
- Figure 1. What is your data source for the track of Typhoon Mangkhut? You haven’t defined this clearly in the figure caption.
- Figure 1. It’s not obvious to me how you’ve defined rapid intensification (RI) in your analysis of Figures 1a and 1b. If RI occurred more than once for a given storm in the Joint Typhoon Warning Center dataset, did you choose the 24-hour period with the biggest increase in maximum sustained wind speed (one period of RI per storm) or include all periods of RI? You need to add these details into the text.
- Figure 2. Why have you chosen to plot the dynamic potential intensity and cooling inhibition index on 7th September 2018, 2 days before Mangkhut reached category 1 intensity? Surely 8th or even 9th September would still be classed as “pre-storm” along most of Mangkhut’s track? For the later part of Mangkhut’s track between 12th to 15th September (furthest west), 7th September becomes less and less representative of the pre-storm environment.
- L93. Does “NWP” refer to western North Pacific (you haven’t spelled out the acronym fully anywhere)? Given how infrequently you use this term, I would remove the acronym elsewhere in the paper too (see general comment about acronyms).
- L103. Be more precise when describing Mangkhut’s evolution (“the area over which it persists” should be replaced with something like “the area over which Mangkhut maintains Category-5 intensity”).
- L118. Spell out “psu” fully (first time you’ve introduced it).
- L129. What is the barrier layer effect for tropical cyclone intensification? A bit more detail here would be useful for the reader.
- L138-140. The implication in this sentence is that a tropical cyclone can only maintain category-5 intensity if passing over an ocean heat content maximum or if warm eddies are present. If this statement is correct, you need to support it more strongly with references from the literature. You do address this point somewhat by exploring some other possible factors later in the section (L149-158), but I think that a stronger link to the literature on the relationship between tropical cyclone intensity and ocean heat

content/warm eddies is required.

- L141-145. I think I know what you're trying to say, but it's difficult for me to visualise fully what you mean by the "positive/negative polarity of transient mesoscale eddies encountered during the passage of TCs". Could you edit the description slightly, or perhaps refer to a figure in another paper?
- L170. Define isopycnals.
- L174. Be more precise with your use of longitude and replace "west of 160°E" with "between 123° and 160°E", to match the description in Figure 3.
- Figure 3. Are the cross-section plots in (a) to (e) valid at a single time, or time-averaged?
- L176-178. Following on from the previous point, you state that the deepening of the thermocline results in the changing ocean heat content values, which suggests that temperature (Fig. 3a) is time-averaged for a single month, in the same way as ocean heat content (Fig. 3f).
- Figure 4 (and related discussion). Be careful about defining August-September as "summer", when it typically refers to June-July-August (same with winter, which usually includes December). At least make the reader aware that you know the difference.
- L200. Does this cooling (0.8 – 0.9°C) correspond to either an area or a length of time?
- L229. Small typo, add "the" before "NEC region".
- L236. Replace "reference" with the paper you're referencing.
- Figure 5. You need to discuss how you calculated the shaded field in Figure 5(c). In the figure caption you have referenced Mangkhut, but in the main text you haven't done this at all. This misalignment makes it difficult for the reader to follow your discussion of the figure.
- L442. Small typo, "comprises of" should be "comprises".
- L465-466. What do you mean by the "common air temperature of hurricanes of 26°C"?
- L465. Replace all instances of "hurricanes" with "tropical cyclones", especially since your analysis doesn't focus on the North Atlantic.
- L491. Replace "TC" with "TCs".
- L516. Replace "typhoon Mangkhut" with "Typhoon Mangkhut" (also L614).
- L528. What's the justification for the decision to calculate the vertical wind shear 5 days prior to the TC arrival? Have previous studies used this value?
- L582. You talk about strong vertical shear "ventilating" the tropical cyclone warm core. Could you be more specific? Is this ventilation the most important effect of vertical shear on tropical cyclone structure?
- L600. How robust is the difference between the salinity during August-September and January-February (Extended Data Fig. 4a-c)? Do these differences (0.5 to 1.0 psu) represent pronounced changes in the ocean salinity?
- L610. Could you be more precise than using "probably" here?

Kang et al., The North Equatorial Current and Rapid Intensification of Super Typhoons

Response to Reviewers

Reviewer #1 (Remarks to the Author):

This study explores the cause of typhoon Mangkhut in 2018. The authors especially emphasize the role of North Equatorial current. They also analyzed the role of ITCZ induced freshening. Overall, this work is interesting. I think most of the analysis focuses on the happening of the long-lasting super typhoon Mangkhut. So, it is more like a case study. But the most intrinsic reason for this is still not clear based on the current evidence. The main reason I think is too much analysis is about the climatology. I will show some examples below. Thus, I think the authors need further investigation into this particular case, that is the typhoon Mangkhut. Please see specific comments below.

Thank you for your comments and your interest in this work. We are aware that the main focus of this article is the super typhoon Mangkhut, and in some ways it is a case study of this TC, but Mangkhut was extraordinary in its formation, and we would suggest that this maybe a precursor for future tropical cyclones. So, from this perspective, Mangkhut is an important milestone in the evolution of TC's. Based on these insightful comments, we have strengthened the analysis of Mangkhut and further related the results to the climatology (see below for details). We respond to each of your specific comments below.

Specific comments:

1. The TC intensification is closely related to the upper ocean heat supply of course, especially for the typhoon Mangkhut, which moves along the NEC region. So, it will be not surprise that the OHC in NEC region plays critically important role in sustaining the strong intensity of Mangkhut. The authors indeed show several largescale conditions that favorable for the intensification, like the large OHC, moderate vertical wind shear and freshwater induced by ITCZ shift. But only Figure 2 is about this particular case. Figures 3 and 4, which is also important for TC intensification, are both on climatological perspective. Although the authors have analyzed in many details, it is still unclear how these climatological variables impact the particular case Mangkhut. For example, in L39-41, the ITCZ northward migration is seasonal march, but why only Mangkhut sustained its high intensity, while other TCs doesn't. This issue exists in the analysis of

precipitation, salinity, buoyancy et al. I suggest the authors to carried out these analyses for Mangkhut as in Figure 2.

We have added three supplementary analyses for Mangkhut in Figure 2 showing the spatial distribution of (a) OHC (b) salinity and (c) salinity-induced N^2 for 9th September 2018, 1 day prior to its rapid intensification. The corresponding figure caption (L607-616) and text (L99-111 and L119-126) are updated in the revised draft.

We have also added in the discussion section further analysis on persistent Super Typhoon Mawar observed in May 2023 in the NEC region (see L220-235 and Supplementary Fig. 8 and 9).

2. The atmosphere changes much faster than the ocean. Thus, I concern that in the analysis of vertical wind shear around L149-158, whether it is still proper to analyze the pre-storm VWS as that of the OHC. The VWS is likely to change with time during Mangkhut. So, how does VWS change during all life cycle of Mangkhut should be investigated.

We have tested different lags in the computation of VWS at each moving center of TC. The results are not very sensitive to the choice of lags between 0 and 2 days for RI and CAT5 phases of Mangkhut (see Supplementary Fig. 2c). Therefore, we have chosen the pre-storm atmospheric conditions 2 days prior to the TC arrival, consistent with the case of the pre-storm upper-ocean parameters. L367-368

3. Figure 6. The authors show the long-term trend of the OHC and TC intensification rate over the NEC region. Actually, many study show that the overall OHC in the WNP has increased not limited to NEC region. So, the figure 6a only shows a part of the overall warming pattern of the WNP. This should be acknowledged in the manuscript and also should be discussed on the potential relationship between NEC warming and the overall warming pattern. Another issue is that ENSO may plays a role in the long-term trend of OHC. Thus, it's signal should be removed.

As responding to editor's comments, the increase in OHC at a rate of $9 \text{ kJ cm}^{-2} \text{ decade}^{-1}$ over the last 4 decades (Fig. 6a) is compared with global increasing rate of OHC. It far exceeds the global ($1.32 \text{ kJ cm}^{-2} \text{ decade}^{-1}$) and all-Tropics ($2.47 \text{ kJ cm}^{-2} \text{ decade}^{-1}$) trends. (L240-242, Supplementary Fig.10) ENSO signals have been removed by subtracting the ENSO-regressed time series from the original time series in the NEC region. The corresponding time series is added in Fig. 6a (solid black line). Note that the long-term trend of OHC does not change with or without ENSO signals. L246-249, L648-651, and Fig.6a

4. The following study is quite relevant to this study, which found that a type of fast decaying El Nino events leads to westward migration of RI events

by affecting the NEC. The authors should not miss it in reviewing the related background: Guo Y. P., and Z. M. Tan, 2018: Westward migration of tropical cyclone rapid-intensification over the Northwestern Pacific during short duration El Nino, Nature Communications, 9, 1507, DOI: 10.1038/s41467-018-03945-y.

We have now cited this article and discussed the results within this context.

5. Figure 6c: The significance. The authors say shading means significant. Does it mean the trends of sea level over the whole Pacific is statistically Significant?

The areas of significant trends are indicated by black dots in Fig. 6c. The figure caption is also reformulated.

Reviewer #2 (Remarks to the Author)

Reviewer: Sam Hardy

Overview

The authors present an interesting and informative analysis of the role of the North Equatorial Current region on the rapid intensification of super typhoons in the western North Pacific. Data are presented impressively both for the case study of Typhoon Mangkhut (2018) and over the longer period of study used for climatological analysis. The interesting findings represent a positive contribution to the field, particularly the analysis on the salinity and ocean stratification, and I congratulate the authors on putting together this piece of work. However, there are several aspects of the discussion that need improving before the paper is suitable for publication, which I have outlined below. My recommendation is Major Revisions.

Thank you for your positive comments and recognition of the contribution of this article to the field of tropical cyclone research.

General comments

- Reduce the number of acronyms throughout the paper. It's difficult for the reader to keep track of them after a certain point.

We have reduced the number of acronyms by omitting 8 acronyms.

- Given the central importance of mesoscale eddies to your analysis, you need to introduce the topic in more detail at the start of the paper. It currently feels like you're assuming that the reader already knows what all these concepts mean.

Thank you very much for pointing this out. We have added the introduction to this topic in more details. L54-56

- Linked to the previous point, how does mesoscale eddy activity contribute to the rapid intensification of tropical cyclones? You need to better introduce the established literature on this topic to give the reader a more solid understanding by the time you get to your specific analysis.

Yes, as above, we have added the related text and references. L56-69

Introduce the "eddy-rich zone" and the "North Equatorial Current" region to the reader in more detail early in the paper and discuss their relevance to tropical cyclone intensification. Much of the analysis that comes later would be easier to follow if the introduction of these terms was more robust.

Yes, very good point and sorry that it was not clear in the earlier version. We have revised this accordingly as well. In addition to the above response for the ERZ, we added a sentence introducing the NEC. L70-72.

- In the abstract you mention the intensification of super typhoons in a warming climate. However, you do not discuss this point any further in the manuscript, only touching on the topic again right at the end of the paper (L248-250). Either expand this part of your analysis or modify the abstract to avoid confusing the reader.

This is a very good point and we decided to modify the abstract to avoid unnecessary emphasis on a warming climate.

- You need more robust discussion of the relationship between upper-ocean stratification and tropical cyclone intensification. Making this change would really add depth to your analysis.

We strengthened the analysis of the Mangkhut case in Fig.2. L99-111

- Create a stronger link between your analysis of Typhoon Mangkhut and your overall conclusions, for example in the Summary and Discussion section. It almost feels like the paper contains two separate bits of analysis, which are not properly linked together. Along the same lines, the paper should end with a more confident and conclusive statement on your findings.

As stated above, we added in Fig. 2 supplementary analysis of the upper-ocean stratification for the Mangkhut case and discussed the latter in relation to the climatology. In the Summary and Discussion section, we also emphasized the link between the analysis of Mangkhut and the overall conclusions. To strengthen the Mangkhut case, we also added the analysis of Mawar (2023), another long-sustained super typhoon in the NEC region recently observed in May 2023. L220-235 and Supplementary Figs. 8 and 9.

Specific comments

- L31-45. Reduce the number of acronyms in the abstract.

Yes, we reduced the number of acronyms, and the following 8 acronyms were omitted: STY (super typhoon), ADT (absolute dynamic topography), SSS (sea surface salinity), 3DPWP (3D Price-Weller-Pinkel), NWP (western North Pacific), NECC (North Equatorial Current), STCC (Subtropical Countercurrent), SLA (sea level anomaly).

L33. Avoid using "this" at the start of the sentence without telling the reader what you're referring to (also L128, L180, L194, L218, L237).

We have replaced "this" by a more suitable word or phrase in each case.

- L35. Define "freshening" here (and again in the main text).

L34. "freshening" is replaced by "freshening in salinity"

L144. "fresher water" is replaced by "fresher (or low-salinity) water"

- L31-45. Discuss how the occurrence of Typhoon Mangkhut relates to your results in a bit more detail. You only mention Mangkhut once (in the first sentence).

At the end of the sentence (L42), we added a phrase: "as clearly demonstrated by Mangkhut."

- L53. Spell acronyms out fully when they occur at the start of a sentence (e.g. "OHC").

Yes, done.

- L56. What do you mean by "warm-core eddies"? You introduce this term in conjunction with a single reference but don't go into any more detail.

We detailed the meaning of "warm-core eddies". L56-57

- L57. Be more specific about "other atmospheric factors".

We specified these factors within the parentheses: "(e.g., vertical wind shear, TC translation speed)". L62-63

- L77. Spell out "NEC" fully; you have introduced the term in the abstract and the caption of Figure 1, but not in the main text.

Yes, done. L71-72

- Figure 1. What is your data source for the track of Typhoon Mangkhut? You haven't defined this clearly in the figure caption.

We reformulated the figure caption, indicating clearly the JTWC data source. L596-597

- Figure 1. It's not obvious to me how you've defined rapid intensification (RI) in your analysis of Figures 1a and 1b. If RI occurred more than once for a given storm in the Joint Typhoon Warning Center dataset, did you choose the 24-hour period with the biggest increase in maximum sustained wind speed (one period of RI per storm) or include all periods of RI? You need to add these details into the text.

For the statistics in (b) and (c), we included all periods (or events) of RI, which is added in the figure caption. L602

- Figure 2. Why have you chosen to plot the dynamic potential intensity and cooling inhibition index on 7th September 2018, 2 days before Mangkhut reached category 1 intensity? Surely 8th or even 9th September would still be classed as "pre-storm" along most

of Mangkhut's track? For the later part of Mangkhut's track between 12th to 15th September (furthest west), 7th September becomes less and less representative of the pre-storm environment.

We agree with your suggestion. We replotted Fig. 2 by changing the pre-storm period from 7th to 9th September and recalculating the corresponding CI index and DPI. L611

- L93. Does "NWP" refer to western North Pacific (you haven't spelled out the acronym fully anywhere)? Given how infrequently you use this term, I would remove the acronym elsewhere in the paper too (see general comment about acronyms).

Yes, it refers to western North Pacific. We removed the acronym of NWP.

- L103. Be more precise when describing Mangkhut's evolution ("the area over which it persists" should be replaced with something like "the area over which Mangkhut maintains Category-5 intensity").

We changed the phrase as suggested. Thank you. L102

- L118. Spell out "psu" fully (first time you've introduced it).

Yes, done. L608, L112-113

- L129. What is the barrier layer effect for tropical cyclone intensification? A bit more detail here would be useful for the reader.

The corresponding sentence is reformulated by adding at its end the following sentence: ", as the increased stratification and stability in the salinity-induced barrier layer reduce the TC-induced vertical mixing and SST cooling." L124-125

- L138-140. The implication in this sentence is that a tropical cyclone can only maintain category-5 intensity if passing over an ocean heat content maximum or if warm eddies are present. If this statement is correct, you need to support it more strongly with references from the literature. You do address this point somewhat by exploring some other possible factors later in the section (L149-158), but I think that a stronger link to the literature on the relationship between tropical cyclone intensity and ocean heat content/warm eddies is required.

At the beginning of the sentence, we added a phrase concerning the tight link between warm eddies and RI of TCs by citing a number of references. L133-134.

- L141-145. I think I know what you're trying to say, but it's difficult for me to visualise fully what you mean by the "positive/negative polarity of transient mesoscale eddies encountered during the passage of TCs". Could you edit the description slightly, or perhaps refer to a figure in another paper?

We reformulated the corresponding sentence. L139-142

- L170. Define isopycnals.

Yes, we added the range of isopycnals (23.0 - 26.5 kg m⁻³). L166

- L174. Be more precise with your use of longitude and replace "west of 160°E" with "between 123° and 160°E", to match the description in Figure 3.

Yes, done. L171

- Figure 3. Are the cross-section plots in (a) to (e) valid at a single time, or time-averaged?

In the figure caption we added the period (1991-2020) over which time-averaging was performed. L622

- L176-178. Following on from the previous point, you state that the deepening of the thermocline results in the changing ocean heat content values, which suggests that temperature (Fig. 3a) is time-averaged for a single month, in the same way as ocean heat content (Fig. 3f).

Figs 3a-e stand for the climatological annual-mean (not monthly mean) sections, while only Fig. 3f shows climatological monthly-mean values for different months. This information is added in the figure caption. L624-625

- Figure 4 (and related discussion). Be careful about defining August-September as "summer", when it typically refers to June-July-August (same with winter, which usually includes December). At least make the reader aware that you know the difference.

We removed "winter/summer" in the figure caption.

- L200. Does this cooling (0.8 - 0.9°C) correspond to either an area or a length of time?

The cited cooling corresponds to the NEC region from the climatology. The corresponding sentence is reformulated. L188-191

- L229. Small typo, add "the" before "NEC region".

Yes, done.

- L236. Replace "reference" with the paper you're referencing.

We reformulated the corresponding sentence. L215

- Figure 5. You need to discuss how you calculated the shaded field in Figure 5(c). In the figure caption you have referenced Mangkhut, but in the main text you haven't done this at all. This misalignment makes it difficult for the reader to follow your discussion of the figure.

The caption is rewritten, indicating also the method of cooling computation for Fig. 5c. In the main text we linked the climatological analysis with Mangkhut analysis. L638-640, L214- 215

- L442. Small typo, "comprises of" should be "comprises".

Yes, corrected. Thank you.

L465-466. What do you mean by the "common air temperature of hurricanes of 26oC"?

The corresponding sentence is reformulated. L301-302

- L465. Replace all instances of "hurricanes" with "tropical cyclones", especially since your analysis doesn't focus on the North Atlantic.

Yes, done.

- L491. Replace "TC" with "TCs".

Yes, done

- L516. Replace "typhoon Mangkhut" with "Typhoon Mangkhut" (also L614).

Yes, done.

- L528. What's the justification for the decision to calculate the vertical wind shear 5 days prior to the TC arrival? Have previous studies used this value?

To be consistent with the pre-storm upper ocean parameters, we changed the pre-storm VWS period from 5 days to 2 days prior to the TC arrival. L367-368. See also the above response to Reviewer 1.

- L582. You talk about strong vertical shear "ventilating" the tropical cyclone warm core. Could you be more specific? Is this ventilation the most important effect of vertical shear on tropical cyclone structure?

The corresponding sentence is removed.

- L600. How robust is the difference between the salinity during August-September and January-February (Extended Data Fig. 4a-c)? Do these differences (0.5 to 1.0 psu) represent pronounced changes in the ocean salinity?

The quoted differences (0.5 to 1.0 psu) are significant and non-negligible in Oceanography because the internationally-required measurement accuracy of salinity is 0.003 psu. In our case, it is the seasonal difference in salinity-induced stratification N2 (Supplementary Fig. 4f) which matters best for the barrier layer effect, as the latter effect controls the TC-induced vertical mixing in the upper layer. We found 10% in difference in CI (cooling inhibition) index due to the seasonal difference in salinity

L610. Could you be more precise than using "probably" here?

The corresponding sentence is reformulated. Supplementary Information L38-44

REVIEWER COMMENTS

Reviewer #1 (Remarks to the Author):

The authors have addressed most of my comments. However, the reply to my first major comments is still not quite adequate. Thus, I still recommend a major revision this round. Please see my specific comments below.

In the last review, I commented that why the favorable climatological environmental condition only makes one TC case, that is Mangkhut, to experience RI. The authors just found another case, the Mawar. I think it still lacks statistical meaning. One good way to makes the arguments that NEC is important to TC RI stronger is to show whether the TCs moving through NEC have much higher probability to experience RI than those that are not. And this difference should also be statistically significant. Only in this way could make the argument more universal.

Reviewer #2 (Remarks to the Author):

You have addressed all my comments thoroughly and professionally and produced a much-improved analysis, which I believe is ready for publication. I only have a couple of very minor comments, which I have listed below. Well done on producing such an interesting and thought-provoking study!

Comments

- L24. I would introduce as 'Super Typhoon Mangkhut' rather than just 'Mangkhut'.
- L78. Does ERZ23 mean the region at 23°N?
- L235. I would be slightly less strong with your wording here. Maybe something like "...supports our new findings from Mangkhut that super typhoons are likely to both rapidly intensify and persist...".

Kang et al., The North Equatorial Current and Rapid Intensification of Super Typhoons

Response to Reviewers

Reviewer #1 (Remarks to the Author):

The authors have addressed most of my comments. However, the reply to my first major comments is still not quite adequate. Thus, I still recommend a major revision this round. Please see my specific comments below.

In the last review, I commented that why the favorable climatological environmental condition only makes one TC case, that is Mangkhut, to experience RI. The authors just found another case, the Mawar. I think it still lacks statistical meaning. One good way to make the arguments that NEC is important to TC RI stronger is to show whether the TCs moving through NEC have much higher probability to experience RI than those that are not. And this difference should also be statistically significant. Only in this way could make the argument more universal.

Thank you for this comment and we apologize for missing this in our initial revision. This is an important point for our article, and we agree that the statistical significance of the NEC for the RI of TCs is critical for the universality of our argument. In order to address this we calculated the statistics for the NEC and non-NEC regions, based on the percentages presented in Fig 1b and 1c. These statistics are presented in supplementary Fig. 1c, and shows that TCs moving through the NEC have a mean probability of 27.6% for experiencing RI compared to 6.1% for those that move through other regions. This difference in probabilities was statistically significant based on the Student's t-test (p -value < 0.001). We added text to reflect this on L85-89.

We hope this addresses your final comment on our manuscript. If however does not satisfy your requirements, please specify how we can address this.

Reviewer #2 (Remarks to the Author):

You have addressed all my comments thoroughly and professionally and produced a much-improved analysis, which I believe is ready for publication. I only have a couple of very minor comments, which I have listed below. Well done on producing such an interesting and thought-provoking study!

Thank you very much for your valuable comments. We think the comments contribute greatly to produce much improved version of the first draft.

Comments

- L24. I would introduce as 'Super Typhoon Mangkhut' rather than just 'Mangkhut'.

Thanks. Agreed. We have added "Super Typhoon" to Mangkhut. L23

- L78. Does ERZ23 mean the region at 23°N?

Thanks. It was corrected as ERZ²³. L79

- L235. I would be slightly less strong with your wording here. Maybe something like "...supports our new findings from Mangkhut that super typhoons are likely to both rapidly intensify and persist...".

Thanks. We revised it with less strong wording. L237

Authors (Remark to the Editor and Reviewers):

Error correction: We found that error number was added to calculate probability of RI event (Fig.6b). The Fig.6b for probability of RI event (red line; %) and their linear regression (thick lines) in the NEC region was corrected (new Fig.6b). L247, L649 (Fig.6b).

REVIEWERS' COMMENTS

Reviewer #1 (Remarks to the Author):

The authors have well addressed my comments. I have no further questions. I recommend an acceptance of this manuscript.